

# Supercurrent-induced Majorana bound states in a planar geometry

**André Melo⋆, Sebastian Rubbert and Anton R. Akhmerov**

Kavli Institute of Nanoscience, Delft University of Technology,
P.O. Box 4056, 2600 GA Delft, The Netherlands

⋆ am@andremelo.org

## Abstract

We propose a new setup for creating Majorana bound states in a two-dimensional electron gas Josephson junction. Our proposal relies exclusively on a supercurrent parallel to the junction as a mechanism of breaking time-reversal symmetry. We show that combined with spin-orbit coupling, supercurrents induce a Zeeman-like spin splitting. Further, we identify a new conserved quantity—charge-momentum parity—that prevents the opening of the topological gap by the supercurrent in a straight Josephson junction. We propose breaking this conservation law by adding a third superconductor, introducing a periodic potential, or making the junction zigzag-shaped. By comparing the topological phase diagrams and practical limitations of these systems we identify the zigzag-shaped junction as the most promising option.


# 1  Introduction

Majorana bound states (MBS) are a promising avenue for fault tolerant quantum computation due to their topological protection [1–4]. While it is possible to realize MBS in spin liquids [5] or in fractional quantum Hall systems [6,7], much of the current experimental effort focuses on systems with induced superconductivity and broken time-reversal symmetry [8–10].

One way of breaking time-reversal symmetry is through an exchange interaction with a ferromagnet [11, 12]. However, in such a setup the interaction is not easily tunable. This creates difficulties in distinguishing MBS from trivial low energy states [13], and makes it necessary to carefully optimize the constituent materials. The most commonly used scheme relies on the Zeeman effect created by an external magnetic field in a proximitized semiconducting nanowire [14–21]. This approach requires strong magnetic fields because the electron spin splitting must exceed the induced superconducting gap in the topological phase. An alternative method relies on the orbital effect of the magnetic field in a three-dimensional geometry, however it also requires strong magnetic fields because of the need to thread a flux comparable to a flux quantum through the device cross-section [22–25]. Magnetic fields suppress the superconducting gap and can create Abrikosov vortices, both detrimental to MBS properties.

Supercurrents also break time-reversal symmetry, and can thus be used to lower the minimal magnetic field required for creating MBS [26,27], or even remove it altogether in hybrid devices combining topological insulators and superconductors [28,29]. Recent proposals have focused on Josephson junctions formed by a two-dimensional electron gas (2DEGs) proximity-coupled to two superconducting terminals [30,31]. In these devices the critical magnetic field reduces significantly when the superconducting electrodes have a phase difference. Such Josephson junctions were realized experimentally [32,33] but a significant critical field reduction is yet to be observed.

Here we propose a setup using a conventional 2DEG and superconducting phase differences to create MBS without an external magnetic field. In order to achieve this, we utilize the idea of Ref. [34], demonstrating that more than two distinct values of superconducting phase are necessary to create a topological phase transition. In particular, we show that applying supercurrents parallel to junction creates a spin splitting that is sufficiently strong to drive a topological phase transition.

# 2  Setup

We consider a 2DEG with spin-orbit interaction covered by two superconductors forming a Josephson junction. The coupling between the superconductor and the semiconductor is strong and therefore the $g$-factor and the spin-orbit coupling are supressed in the covered regions [35]. The superconductors carry supercurrents in opposite directions along the junction (Fig. 1). We model this system using an effective 2-dimensional Hamiltonian combining parabolic dispersion and Rashba spin-orbit interaction:

$$H = \left( \frac{p_x^2 + p_y^2}{2m} - \mu \right) \sigma_0 \tau_z + \xi(y) \alpha (p_x \sigma_y - p_y \sigma_x) \tau_z + \mathrm{Re}\Delta(x,y) \sigma_0 \tau_x + \mathrm{Im}\Delta(x,y) \sigma_0 \tau_y, \quad (1)$$

where $p_{x,y} = -i\hbar\partial_{x,y}$, $m$ is the effective electron mass, $\mu$ the chemical potential, $\alpha$ the Rashba spin-orbit interaction strength and $\Delta(x,y)$ the superconducting gap. The indicator function $\xi(y) = 0$ under the superconductor and $\xi(y) = 1$ otherwise. Finally, $\sigma_i$ and $\tau_i$ are the Pauli matrices in the spin and the electron-hole space. This Hamiltonian has a particle-hole symmetry $\mathcal{P} = \tau_y \sigma_y K$, with $K$ complex conjugation. Because the superconductors carry a supercur-

rent, their phase depends linearly on $x$:

$$\Delta(x,y) = \begin{cases} \Delta_0 \exp(2\pi i x/\lambda_T) & W/2 < y < W/2 + L_{\mathrm{sc}}, \\ 0 & |y| < W/2, \\ \Delta_0 \exp(-2\pi i x/\lambda_B) & -W/2 - L_{\mathrm{sc}} < y < -W/2, \end{cases} \tag{2}$$

with $W$ the width of the Josephson junction, $\lambda_T$ and $\lambda_B$ the winding lengths of the super-conducting phase in the two superconductors, and $\Delta_0$ the magnitude of the induced super-conducting gap. Making the superconducting phase depend only $y$ coordinate coordinate is insufficient, because at $k_x = 0$ the spin-orbit coupling may be removed by a transformation $\psi(y) \to \exp[i\sigma_x f(y)]\psi(y)$, and therefore all states are doubly degenerate. This degeneracy was overlooked in Ref. [22] when analyzing the effective two-dimensional Hamiltonian of the semiconducting slab.

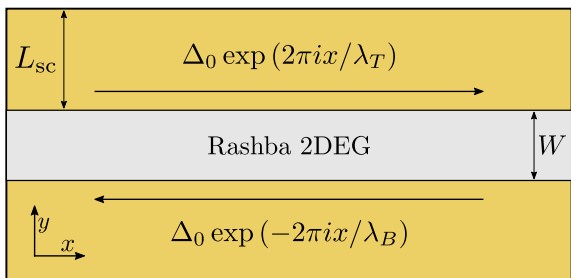

Figure 1: A 2DEG with Rasba spin-orbit coupling covered by two conventional su-perconductors. The superconductors carry longitudinal supercurrents in opposite directions, indicated by the horizontal arrows.

To characterize the topological properties of the setup we apply the finite difference ap-proximation to the continuum Hamiltonian Eq. (1) with a lattice constant $a = 10\,\mathrm{nm}$, and numerically study the resulting tight-binding Hamiltonian using the Kwant software pack-age [36]. We use the implementation of Ref. [37] as a starting point. Whenever necessary we use Adaptive [38] to efficiently sample the parameter space. The source code and data used to produce the figures in this work are available in Ref. [39].

## 3 Creating a topological phase

We illustrate the appearance of the topological phase by introducing the necessary ingredients one by one. The resulting band structures are computed through sparse diagonalization of the Hamiltonian for several values of the Bloch wave vector $\kappa$ corresponding to the super-cell of the device. We choose the following parameter values, unless specified otherwise. The effective electron mass is $m = 0.04 m_e$, with $m_e$ the free electron mass, $\lambda_T = \lambda_B = \lambda = 370\,\mathrm{nm}$, $\Delta_0 = 1\,\mathrm{meV}$, $\alpha = 10\,\mathrm{meV\,nm}$, as well as the geometrical parameters $L_{\mathrm{sc}} = 200\,\mathrm{nm}$, $W = 150\,\mathrm{nm}$.

### 3.1 Phase winding and inversion symmetry

We observe that the band structure in presence of phase winding has a spin splitting at $\kappa = 0$, as shown in Fig. 2. The level crossing at $\kappa = 0$ may be protected only by the Kramers degeneracy appearing when $H$ commutes with an antiunitary operator squaring to $-1$. In absence of winding, this condition is fulfilled by the time-reversal symmetry $\mathcal{T} = \sigma_y K$. We identify that even in presence of winding, the Hamiltonian commutes with the operator $\delta(y + y')\mathcal{T}$, except for the transverse spin-orbit coupling $\alpha p_y \sigma_x$. Therefore the avoided crossing is produced

by a combination of the winding and of the transverse spin-orbit coupling breaking all the remaining time-reversal-like symmetries of the system. In Fig. 2 we also demonstrate that removing the transverse spin-orbit coupling restores the degeneracy of levels at $\kappa = 0$. We conclude that the width $W$ of the normal region must be comparable to the spin-orbit length $l_{so} = \hbar/m\alpha$ in order for the transverse spin-orbit to have a sufficient impact and to cause a spin splitting. The level crossings at $\kappa = \pi$ stay protected by a nonsymmorphic antiunitary symmetry with an operator $\tau_z \delta(y + y')\delta(x - x' + \lambda/2)K$.

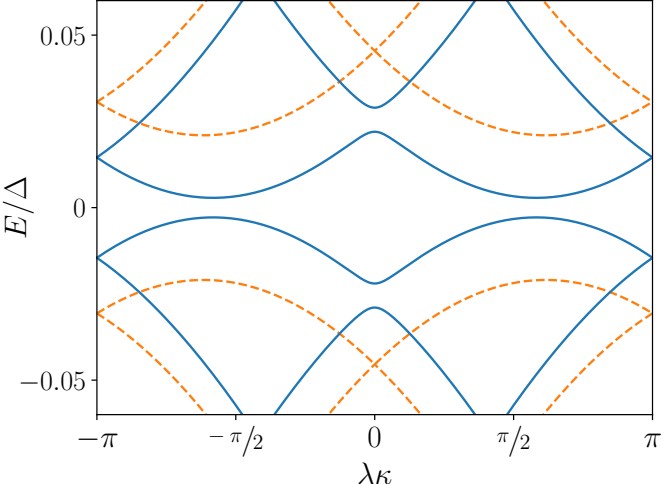

Figure 2: Band structures of systems with spin-orbit interaction at $\mu = 0.17$ meV. The avoided level crossings at $\kappa = 0$ are a consequence of an effective Zeeman interaction originating from the combination of the spin-orbit coupling and the supercurrents carried by the superconductors. Removing transverse spin-orbit coupling restores Kramer's deneneracy at $\kappa = 0$ and results in the band structure plotted with dashed lines.

Furthermore, we see that the spectrum is reflection symmetric about $\kappa = 0$. This is a consequence of the inversion symmetry of the Hamiltonian $[H, I] = 0$, with the inversion symmetry operator $I = \delta(x + x')\delta(y + y')\sigma_z$. Since choosing $\lambda_T \neq \lambda_B$ breaks the inversion symmetry, it may close the band gap at finite momentum, as illustrated in Fig. 3, where we chose $\lambda_T = 2\lambda_B = 700$ nm and $\mu = 0.42$ meV. Preserving inversion symmetry therefore maximizes the parameter range supporting gapped spectra.

## 3.2 Breaking the charge-momentum conservation law

The band structure in Fig. 2 resembles that of a proximitized nanowire with spin-orbit interaction and Zeeman field [14, 15]. By analogy it is then natural to expect that tuning the chemical potential such that the two spin states at $\kappa = 0$ have opposite energies should result in a topologically nontrivial band structure. Instead we observe a gapless band structure with band gap closings at finite $\kappa$, as shown in Fig. 4(a).

The crossings in the spectrum are protected because every Andreev reflection in this setup is accompanied by a wave vector change of $\pm 2\pi/\lambda$. Therefore the Hamiltonian conserves the charge-momentum parity

$$\mathcal{O} = (-1)^n \tau_z, \quad [H, \mathcal{O}] = 0. \tag{3}$$

Here $n \equiv \lambda(k_x - \kappa)/2\pi$ is the number of the unit cell in reciprocal space. We visualize this conservation law in Fig. 5. Because $\{\mathcal{P}, \mathcal{O}\} = 0$, each eigenstate $|\Psi\rangle$ of the Hamiltonian with

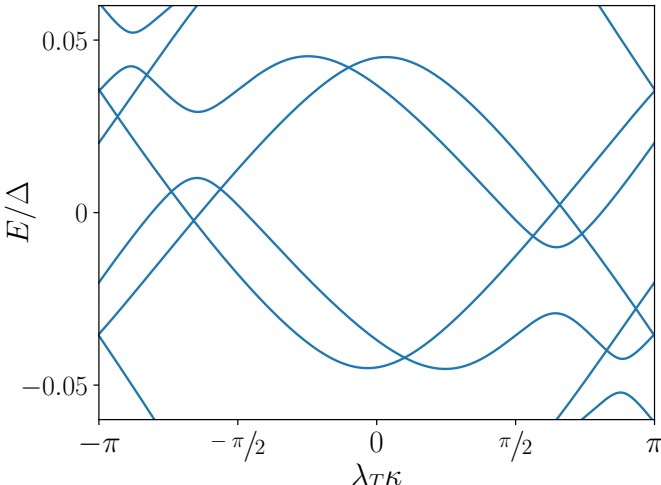

Figure 3: Gapless bandstructure due to broken inversion symmetry resulting from different supercurrent densities ($\lambda_T = 2\lambda_B$).

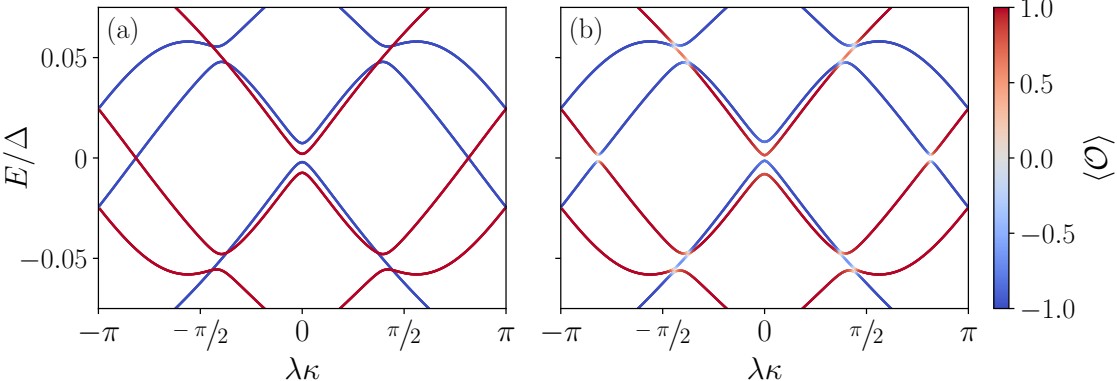

Figure 4: Band structures of the system with the Fermi level tuned inside the avoided crossing at $\kappa = 0$ with (a) a zero band gap due to charge-momentum parity conservation, and (b) finite band gap due to a periodic potential. The bands are colored according to the expectation value of $\mathcal{O}$.

energy $E$, Bloch wave vector $\kappa$, and charge-momentum parity $\mathcal{O}$ has a partner $\mathcal{P}|\Psi\rangle$ with $-E$, $-\kappa$, and $-\mathcal{O}$. Topological phase transitions occur whenever such a pair of states crosses zero energy at $\kappa = 0$ or $\kappa = \pi$. As a consequence, in the topological regime the difference of the number of states with positive $E$ and $\mathcal{O}$ at $\kappa = 0$ and those at $\kappa = \pi$ is odd. Therefore the topological phase requires at least one band with positive $\mathcal{O}$ (and its particle-hole symmetric partner with negative $\mathcal{O}$) to cross zero energy between $\kappa = 0$ and $\kappa = \pi$. This prohibits a gapped topological phase as long as $\mathcal{O}$ is conserved.

Since a gap is necessary for topologically protected MBS, we consider the following strategies for breaking the charge-momentum parity conservation:

- adding a periodic potential

$$\delta V = V \cos(2\pi x/\lambda_V)\sigma_0 \tau_z, \tag{4}$$

with $V$ the amplitude of the potential and $\lambda_V$ its periodicity;

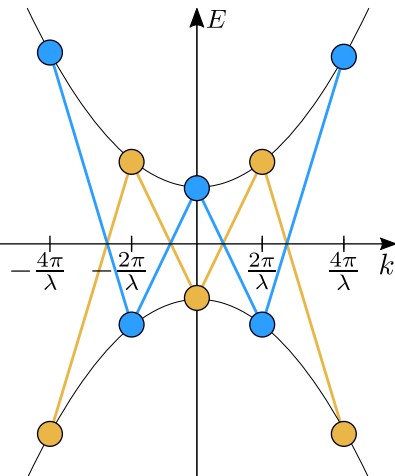

Figure 5: Schematic normal state band structure of the Hamiltonian of Eq. (1). For illustration purposes we neglect the spin-orbit coupling. The dots represent momentum eigenstates with Bloch momentum $\kappa = 0$ and the lines denote couplings introduced by the superconductors. The colors correspond to different eigenvalues of the charge-momentum parity.

- adding an extra superconductor in the middle, as sketched in Fig. 6 (a), so that $\Delta(x, y)$ becomes:

$$\Delta(x, y) = \begin{cases} \Delta_0 \exp\left(-2\pi i x / \lambda\right) & y > W/2, \\ \Delta' & w/2 > |y|, \\ 0 & w/2 < |y| < W/2, \\ \Delta_0 \exp\left(2\pi i x / \lambda\right) & y < -W/2, \end{cases} \tag{5}$$

where $w$ is the width of the middle superconductor and $\Delta'$ its superconducting gap;

- adding a zigzag modulation to the junction shape [40] with period $z_x$ and amplitude $z_y$, as depicted in Fig. 6(b).

These modifications couple the eigensubpaces of $\mathcal{O}$ as shown in Fig. 7 and open a gap in the topological regime. We verify that this is the case by adding a periodic potential with $V = 0.005$ meV and $\lambda_V = \lambda$, which results in a gapped topologically-nontrivial band structure shown in Fig. 4(b).

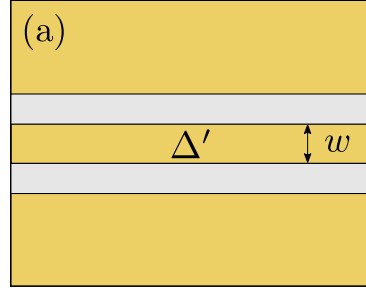
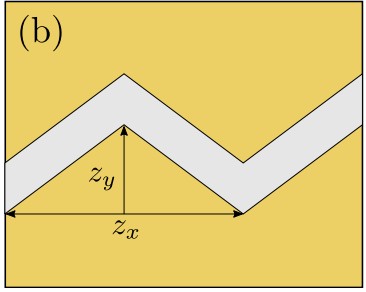

Figure 6: Schematics of systems with broken charge-momentum parity symmetry due to (a) a third superconductor carrying no supercurrent, and (b) a zigzag-shaped junction.

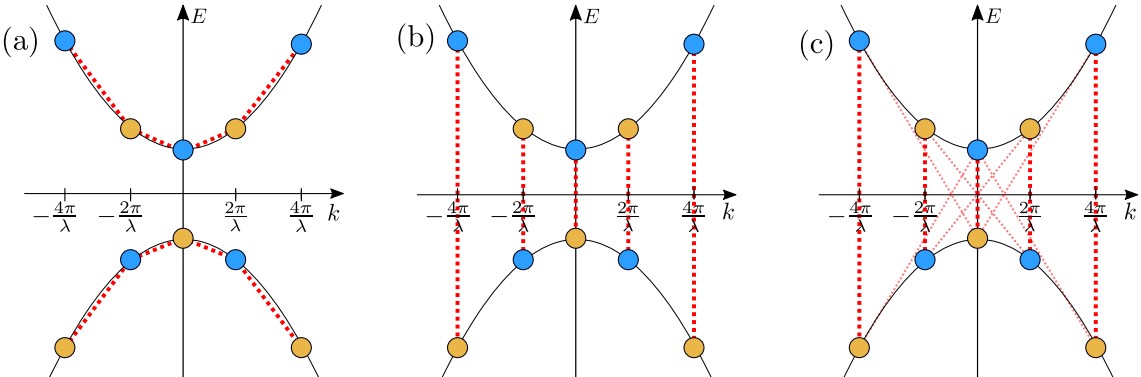

Figure 7: Schematic band structure of the Hamiltonian of Eq. (1) with charge-momentum parity breaking terms. The red dashed lines denote the symmetry breaking couplings introduced by (a) a periodic potential, and (b) a third superconductor, and (c) zigzag-shaped junction. By projecting the zizag junction Hamiltonian onto a plane wave basis we have verified that it introduces couplings to higher harmonics [39] which we denote with narrower transparent lines.

## 4   Phase diagrams

In order to check how robust the resulting topological superconductivity is, we study the topological phase diagrams of the three candidate systems as a function of $\lambda$ and $\mu$, focusing especially on the effect of winding of the superconducting phase becoming incommensurate with the other periods appearing in the Hamiltonian: $\lambda_V$ and $z_x$. For illustration purposes we choose the parameters $\alpha = 20\,\text{meV}\,\text{nm}$, $z_x = 515\,\text{nm}$, $z_y = 37.5\,\text{nm}$, $V = 0.15\,\text{meV}$, $\lambda_V = 515\,\text{nm}$, $\Delta' = \Delta_0 = 1\,\text{meV}$ and $w = 10\,\text{nm}$. Because our goal is a qualitative exploration of the topological phase diagram we neglect the impact of the zigzag shape on the phase winding pattern. This is also a good approximation because the zigzag modulation is small ($z_x \sim 10 \times z_y$). We utilize the scattering formalism to construct the topological phase diagram when the winding length $\lambda$ of the superconducting phase is incommensurate with the periodicity of the potential $\lambda_V$ or the period of the zigzag modulation $z_x$. Specifically, we construct a finite but large system with length $L_x = 10.3\,\mu\text{m} = 20z_x$ with two normal leads attached, shown in Fig. 8(a). We then compute the scattering matrix as a function of energy and compute the topological invariant $Q = \text{sign}\det r$, where $r$ is the reflection block of the scattering matrix [41]. We estimate the gap as the lowest energy at which the total transmission between two leads $T_{12} = 1/2$, as illustrated in Fig. 8(b).

Because adding a third superconductor preserves inversion symmetry regardless of $\lambda$, the phase diagram of the system with 3 superconductors is gapped except for phase transitions. In contrast, the periodic potential and zigzag systems are only inversion symmetric when the periods of different Hamiltonian terms are equal, that is when $\lambda = \lambda_V$ and $\lambda = z_x$. Once parameters become incommensurate the gap closes quickly and the diagrams have large gapless regions. However, the topological phase of the system zigzag geometry is significantly more robust to incommensurate parameters than that of the periodic potential and tolerates variations of $\lambda$ of approximately 10%. We also observe that the zigzag geometry is sufficiently robust to support a gapped topological phase with only one superconductor, see App. A.

The shape of the topological regions has a complex dependence on $\mu$ and $\lambda$ that does not seem amenable to analytical treatment. Additionally, the topological gap is smaller than the full superconducting gap by approximately a factor of 50, likely due to a suboptimal choice of parameters, rather than a fundamental limitation of the setups.

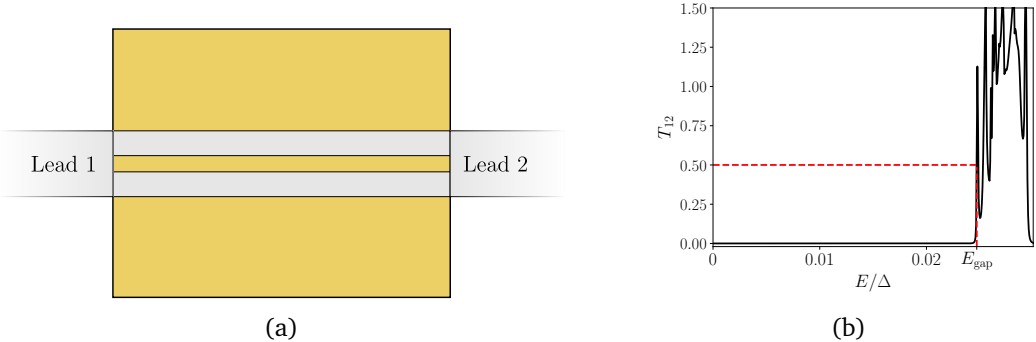

Figure 8: To compute the band gap we (a) attach two two normal leads to the system and (b) compute the transmission between the leads; we then approximate the gap to be the energy at which transmission exceeds 0.5.

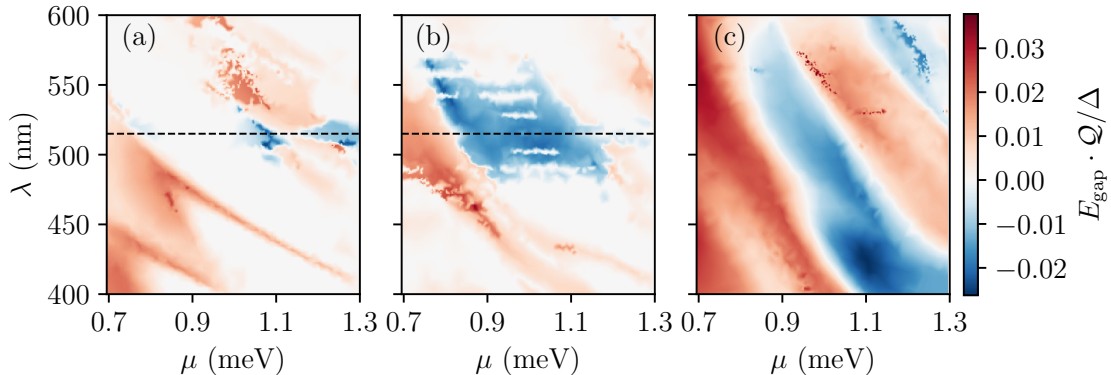

Figure 9: Phase diagrams for systems with (a) a periodic potential, (b) a zigzag-shaped junction, and (c) a third superconductor carrying no supercurrent. The dashed line indicate the region where the systems have commensurate parameters, that is $\lambda = \lambda_V$, and $\lambda = z_x$. Negative values correspond to topologically non-trivial systems.

## 5  Summary

In summary, we have shown that the winding of a superconducting phase is a sufficient source of time-reversal symmetry breaking to create MBS in Josephson junctions. By performing symmetry analysis we have identified the breaking of the charge-momentum parity conservation law as the key ingredient for tuning the system into a gapped topological regime. Furthermore, we showed that preserving inversion symmetry maximizes the size of the parameter regions supporting gapped spectra.

The only magnetic field in the system is caused by the supercurrents in the electrodes. To estimate the magnitude of the magnetic field we approximate the supercurrents and the resulting magnetic field through the relations $I = hdW/(\lambda_L^2 \mu_0 2e\lambda)$ and $B = \frac{\mu_0 I}{2\pi W}$, where $d$ is the thickness of the superconductor, $\lambda_L$ the London penetration depth, and $\mu_0$ the vacuum permeability. Using experimentally realistic values of $d = 10\,\text{nm}$, $\lambda_L = 200\,\text{nm}$ (niobium) and $\lambda = 250\,\text{nm}$ yields $\sim 0.3\,\text{mA}$ and $B \sim 0.2\,\text{mT}$, which is negligible in a mesoscopic superconductor.

The periodic potential scheme is the most challenging to implement experimentally, since it requires patterning a large number of gates. Additionally this scheme requires almost ex-

actly commensurate $\lambda$ and $\lambda_V$. Adding a third superconductor has the advantage of preserving inversion symmetry regardless of the phase winding length $\lambda$. On the other hand it is sensitive to the geometry: the width of the middle strip $w$ must be large enough to allow Andreev reflections, but shorter than the superconducting coherence length in order to allow transmission between the top and bottom superconductors. The zigzag-shaped junction has a larger tolerance to incommensurate parameters compared to the periodic potential and is less sensitive to the details of the geometry than the third superconductor. Furthermore, it can be be fabricated with current techniques [42], making it the most promising scheme.

We have excluded the effects of disorder and aperiodic variations in the geometry or the electrostatic environment of the device. Such perturbations destroy translation symmetry and couple states with different Bloch momenta, thus also breaking the charge-momentum parity, and potentially offering a simpler approach to creating a topological phase. Another direction of further research would is to identify the system geometry and parameters maximizing the topological gap of the systems.

## Acknowledgements

We are grateful to A. Beukman, B. van Heck, T. Laeven, A. Stern, and D. Sticlet for useful discussions, and K. Pöyhönen and D. Varjas for identifying the symmetry that protects the degeneracy at $\kappa = \pi$ in the absence of transverse spin-orbit interaction.

**Author contributions** A. A. formulated the project goal and oversaw the project. S. R. implemented the initial version of the simulation, identified the role of symmetries and the charge-momentum parity and analysed the commensurate regime of the 3 superconductor and periodic potential systems. A. M. implemented the final version of the code, analyzed the topological phase diagrams, and produced the publication figures. All authors made significant contributions to the planning of the project and writing the manuscript.

**Funding information** This work was supported by the Netherlands Organization for Scientific Research (NWO/OCW), as part of the Frontiers of Nanoscience program, an NWO VIDI grant 016.Vidi.189.180, and an ERC Starting Grant STATOPINS 638760.

## A  System with a single zigzag-shaped superconductor

A system with a single zigzag-shaped superconducting contact, as shown in Fig. 10, may still support a topological phase despite having strongly broken inversion symmetry. In Fig. 11 we show a phase diagram for such a system with $z_x = 360\,\text{nm}$, $z_y = 75\,\text{nm}$, and $L_x = 7.2\,\mu\text{m} = 20 z_x$.

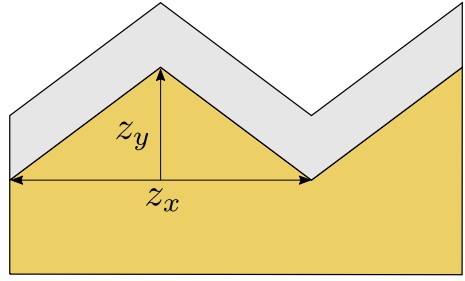

Figure 10: Schematic of a system with a single zigzag-shaped superconductor.

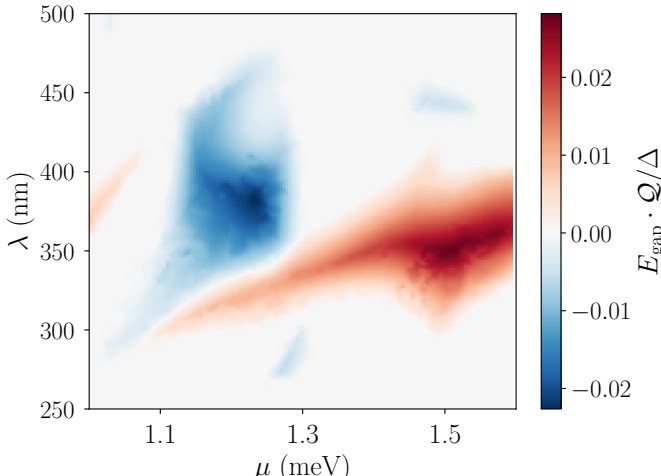

Figure 11: Phase diagram of a system with a single superconductor in a zigzag shape.

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
