# Peer review of "Supercurrent-induced Majorana bound states in a planar geometry"

_SciPost Physics, doi:SciPost Phys. 7, 039 (2019)_

## Round 2 · Referee Report · Anonymous (Referee 1) · 2019-7-2

Strengths

  1. Relates to a field of high current experimental and theoretical interest
  2. Offers a new way to create a topological superconductor
  3. Is written in a way that is accessible and useful also for experimentalists, and might inspire future experimental works
  4. Uses an open source software and provides links to codes and parameters used in the paper

Weaknesses

  1. The numerics are not always supported with analytical or intuitive insights
  2. The dependence of the results on the various different parameters remains unclear
  3. The predicted superconducting gaps in the proposed systems are very small, which seriously limits the experimental relevance

Report

The paper "Supercurrent-induced Majorana bound states in a planar geometry" is a theoretical proposal for realizing Majorana bound states (MBS) in planar Josephson junction devices, which relies entirely on using a supercurrent to break time-reversal symmetry (meaning that no external magnetic field is needed). The authors use a rather simple 2D model of a 2DEG partly covered by a superconductor and then discretize this and solve it as a tight-binding problem. They find that in the simplest geometry there is no gapped topological phase, and then investigate three different ways (adding a third superconductor, adding a periodic potential or making the Josephson junction zagzag-shaped) to break the symmetry and induce a topological phase. They scan the parameter space and calculate where a topological phase can be expected and give an estimate of the superconducting gap in the topological regions.

In short, I believe that the results are scientifically sound, the paper is well-written, and the work represents a substantial advance in a very active scientific field. The strengths and weaknesses, according to my opinion, are given above. In addition to these points, I have a few technical questions:

  1. From the text after Eq. 1, I get the impression that the spin-orbit coupling is taken to be zero in the regions with direct superconducting proximity effect. Why? Would this region not just be the same 2DEG as elsewhere, but covered with a superconducting material?

  2. On page 7, the authors say that they assume the same phase winding in the zigzag system as in the straight one. I guess this would be the result if the supercurrent density is unaffected by the zigzag pattern. Why, and under which circumstances, do the authors expect this to be a good approximation?

  3. Related to weakness 3. above, on page 8 the authors write that the small superconducting gap is "likely due to a suboptimal choice of parameters". Why do they think so?

  4. (Minor point) Why is the induced superconducting gap taken to be 1 meV? This seems rather large. Especially considering that they authors use the London penetration depth for Al in their estimate on page 9, and Delta for Al is much smaller.

Requested changes

I think that my questions 1-4 in the report should be addressed and probably warrants some modifications to the paper. In addition, I think the paper would be improved by addressing the weaknesses 1-3 above. In particular, I think the paper would be greatly improved by additional physical (analytical and/or intuitive) insight into how the result depends on the different parameters. This might also hint at how the central problem for an experimental realization (the small gap) can be solved.

  • validity: top
  • significance: good
  • originality: good
  • clarity: high
  • formatting: excellent
  • grammar: excellent

Author:  André Melo  on 2019-09-11  [id 600]

(in reply to Report 1 on 2019-07-02)

We thank the referee for the time and effort in reviewing our manuscript. Below we address the technical questions posed in the report.

  1. From the text after Eq. 1, I get the impression that the spin-orbit coupling is taken to be zero in the regions with direct superconducting proximity effect. Why? Would this region not just be the same 2DEG as elsewhere, but covered with a superconducting material?

The coupling between the superconductor and the semiconductor in the covered regions is strong, which suppresses spin-orbit coupling and the $g$-factor. To clarify this point we have added a remark and reference in the updated manuscript.

  1. On page 7, the authors say that they assume the same phase winding in the zigzag system as in the straight one. I guess this would be the result if the supercurrent density is unaffected by the zigzag pattern. Why, and under which circumstances, do the authors expect this to be a good approximation?

The zigzag amplitude in the geometries we consider is small ($z_x ∼ 10 z_y$). Under these conditions we expect the corrections to the supercurrent distribution to be negligible. Additionally, the supercurrents in the system act merely as a source of time-reversal symmetry breaking. Therefore, our qualitative results are independent of the details of their spatial distribution. We have revised the manuscript to clarify this point.

  1. Related to weakness 3. above, on page 8 the authors write that the small superconducting gap is "likely due to a suboptimal choice of parameters". Why do they think so?

There is no fundamental theoretical reason that forbids larger topological gaps in the system (e.g. vortices on top of a topological insulator have a topological gap equal to $\Delta$). Furthermore, we did not attempt to optimize the current parameters: they were chosen merely for illustration purposes. We are therefore convinced that a more exhaustive parameter search would reveal parameter regions with larger gaps. Because such a parameter search would amount to a significant additional numerical work, we have chosen to focus on the prototype implementation.

  1. (Minor point) Why is the induced superconducting gap taken to be 1 meV? This seems rather large. Especially considering that they authors use the London penetration depth for Al in their estimate on page 9, and Delta for Al is much smaller.

We thank the referee for pointing out this discrepancy in the system parameters. Superconductor-semiconductor heterostructures with Nb can have induced gaps of the order 1 meV (see for example https://www.nature.com/articles/s41565-017-0032-8?WT.feed_name=subjects_materials-science). For consistency we have changed the penetration depth to that of Nb thin films.

The referee also points out that

(...) the paper would be greatly improved by additional physical (analytical and/or intuitive) insight into how the result depends on the different parameters. This might also hint at how the central problem for an experimental realization (the small gap) can be solved.

We agree with the referee's evaluation. However, given the large number of system parameters and high cost of the necessary simulations, such an investigation is highly complex and we have chosen to limit the scope of the current manuscript.

We hope that our revision adequately addresses the issues pointed out by the referee. We look forward to hearing from you.

Best regards, André Melo Sebastian Rubbert Anton Akhmerov

---

## Round 3 · Author Response

Minor changes to address the referee report. See list of changes and referee reply for details.

---

## Round 3 · List of Changes

• Added remark and reference to explain why we take spin-orbit coupling to be zero in the proximitized regions.
  • Added explanation of why our qualitative results are unaffected by the details of the supercurrent distribution in the zigzag geometry.
  • Changed penetration depth from Al to Nb in the magnetic field estimate.

---

## Editorial Decision

published